# Risk Factors for Anemia and Micronutrient Deficiencies among Women of Reproductive Age—The Impact of the Wheat Flour Fortification Program in Uzbekistan

**DOI:** 10.3390/nu12030714

**Published:** 2020-03-07

**Authors:** Nicolai Petry, Fakhriddin Nizamov, Bradley A. Woodruff, Regina Ishmakova, Jasur Komilov, Rita Wegmüller, James P. Wirth, Diyora Arifdjanova, Sufang Guo, Fabian Rohner

**Affiliations:** 1GroundWork, 7306 Fläsch, Switzerland; woody@groundworkhealth.org (B.A.W.); rita@groundworkhealth.org (R.W.); james@groundworkhealth.org (J.P.W.); fabian@groundworkhealth.org (F.R.); 2UNICEF, Tashkent 100029, Uzbekistan; fnizamov@unicef.org (F.N.); darifdjanova67@mail.ru (D.A.); sguo@unicef.org (S.G.); 3Vitamed Clinic LLC, Tashkent 100059, Uzbekistan; lab@vitamed.uz; 4MedStandart LLC, Tashkent 100034, Uzbekistan; jasurkomilov@gmail.com

**Keywords:** wheat flour fortification, women of reproductive age, folic acid, iron, iron deficiency, vitamin A insufficiency, folate deficiency, anemia, Uzbekistan

## Abstract

Food fortification can be effective in reducing the prevalence of anemia and micronutrient deficiencies. This study assessed risk factors for—and the impact of the wheat flour program in Uzbekistan on—anemia, and iron and folate deficiency (FD) in non-pregnant women (NPW) of reproductive age. National data were analyzed for risk factors using multivariable regression. Additional iron intake from fortified flour was not associated with iron deficiency (ID) and did not result in a significantly different prevalence of anemia regardless of the levels, whereas women with additional folic acid intake had a lower relative risk (RR) of FD (RR: 0.67 [95% CI: 0.53, 0.85]). RR for anemia was greater in women with ID (RR: 4.7; 95% CI: 3.5, 6.5) and vitamin A insufficiency (VAI; RR 1.5; 95% CI: 1.3, 1.9). VAI (RR: 1.4 [95% CI: 1.3, 1.6]) and breastfeeding (RR: 1.1 [95% CI: 0.99, 1.2]) were associated with increased risk of ID, while being underweight reduced the risk (RR: 0.74 [95% CI: 0.58, 0.96]). Breastfeeding (RR: 1.2 [95% CI: 1.1, 1.4]) and inflammation (RR: 1.2 [95% CI: 1.0, 1.3]) increased risk of FD. FD results indicate that the fortification program had potential for impact, but requires higher coverage of adequately fortified wheat flour and a more bioavailable iron fortificant.

## 1. Introduction

Micronutrient deficiencies are widespread and estimated to affect more than 2 billion people globally, significantly contributing to the global burden of disease [1]. The extent of the most common deficiencies such as iron, vitamin A and iodine has been well described, but for many less frequently occurring deficiencies, data are scarce. For example, nationally representative prevalence estimates for folate deficiency (FD) are rare, especially for low-income countries [2]. However, given the role of folate in the development of neural tube defects (NTD) and megaloblastic anemia [3], FD merits further investigation. Folic acid supplementation has been successful in increasing blood folate levels and reducing NTD [4], but the typical approach of providing iron folic acid supplements during antenatal visits is limited because many women do not start consuming folic acid supplements until too late in pregnancy, when the embryo’s neural tube has already been closed [5,6]. Food fortification has been shown to be effective in increasing blood folate levels and reducing NTD [7,8], but mixed results have been observed for iron deficiency (ID) and anemia [9], which could be explained by the choice of poorly bioavailable iron fortificants as well as a high consumption of iron-absorption inhibitors [10] or physiological conditions such as inflammation, which prevent iron absorption [11].

A nationally representative nutrition survey in 2017 reported that almost 50% of non-pregnant women (NPW) in Uzbekistan had FD [12], compared to less than 30% in 2008 [13], while ID and anemia prevalence were 20% and 41% in 2017, respectively, and remained almost unchanged during the same time period. A potential explanation for the significant increase in FD could be the decreasing coverage with fortified flour from 71% in 2008 to only 30% in 2017. It remains unclear, however, why this did not affect prevalence of ID or anemia. In Uzbekistan, wheat flour fortification started in 2001 and was scaled-up in 2005 by mandating fortification with electrolytic iron (40–65 ppm), folic acid (1–4 ppm), zinc (15–30 ppm), thiamin (1.6–10 ppm), riboflavin (1.4–12 ppm), and niacinamide (10–40 ppm) in state-owned mills [14]. Despite the additional mandate to fortify wheat flour in all mills in Uzbekistan, the premix remained unchanged until the implementation of the Uzbekistan Nutrition Survey (UNS) in 2017. Thereafter, vitamin B12 was added to the premix and the iron fortificant was changed from electrolytic iron to NaFeEDTA at a level of 16–40 ppm [15]. 

The objective of this analysis was to identify risk factors for anemia, ID, and FD, and to investigate whether or not the consumption of fortified wheat flour prevented the development of anemia, ID, or FD among NPW 15–49 years old.

## 2. Materials and Methods 

### 2.1. Survey Design and Participants

The UNS was a household-based survey conducted in October and November 2017. Two-stage sampling was carried out. In the first stage, makhallas, the smallest administrative unit, served as the primary sampling unit. In the second stage, households served as the basic sampling unit. A more complete description of survey procedures has been provided elsewhere [12]. In brief, NPW 15 to 49 years of age, pregnant women, and children less than five years of age were recruited as individual participants in the survey. Of those, only NPW (both lactating and non-lactating) were included in the analysis presented in this manuscript. NPW were recruited only from every second selected household. An oblast, the equivalent of a ‘region’ in Uzbekistan, served as the unit of stratification. Within each of the 14 strata, 25 primary sampling units were selected with probability proportional to population size. To ensure the most accurate sampling frame for the selection of households, dedicated teams conducted door-to-door visits to update household lists for each selected cluster a few weeks prior to survey data collection. In each selected cluster, 12 households were selected at random with equal probability, resulting in a target sample size of 4200 households. 

### 2.2. Data Collection and Laboratory Analysis

Prior to data collection, experienced field workers received intensive training consisting of classroom instruction and practice, laboratory practice, and field testing of all survey procedures. Training for phlebotomists included blood collection techniques, labeling of samples, and maintenance of the cold chain for transporting blood specimens. Tablet computers were used for direct data entry during interviews. Skip patterns were built into the electronic questionnaires, which sped up interviewing as well as minimized erroneous entries. Household and individual questionnaires were available in English, Uzbek, Russian, and Karakalpak. Interviews were conducted in the interviewee’s preferred language.

A short household questionnaire was administered to the head of the household or, if this person was not present, to another knowledgeable adult household member. The household questionnaire contained modules collecting data on demographic and socio-economic characteristics, household composition, and household food purchase and consumption. At each household, a flour sample (approximately 50 g) was requested, which was collected and stored in airtight containers for later analysis. Short questionnaires were administered to all recruited women, collecting information on minimum dietary diversity (24 hour recall), consumption of and knowledge about fortified and fortifiable foods, consumption of vitamin and mineral supplements, and demographic factors. 

Anthropometry was conducted in all participating NPW using standard methods [16] on a SECA (Hamburg, Germany) scale (UNICEF, #S0141021) and a standard wooden height board (UNICEF, #S0114540). Blood was collected from all NPW via venipuncture into 6-mL serum tubes (Becton Dickinson, Franklin Lakes, NJ, USA). Using a DIFF-Safe device (Becton Dickinson, Franklin Lakes, NJ, USA), a small amount of blood was extracted from the tubes onto a weighing boat to assess hemoglobin concentration using a portable hemoglobinometer (Hb301+, HemoCue, Angelsholm, Sweden). Quality control (QC) of the HemoCue devices was conducted and recorded daily using control materials (Eurotrol, Ede, Netherlands). Remaining whole blood was placed in a cool box containing cold packs to ensure storage at about 4 °C in the dark until further processing later the same day. 

Every evening, blood and flour samples were transported to the regional blood transfusion centers for centrifugation and aliquotation of serum into cryovials appropriately labeled with the respondents’ identification numbers. Upon completion of field work in all regions, serum samples were collected and transported frozen to Tashkent, where they were stored at −80 °C until analysis. In the same transport, flour samples were taken to the ‘Donmahsulotlari LLC’ laboratory in Tashkent.

Sera were analyzed for ferritin, C-reactive protein (CRP), alpha1-acid-glycoprotein (AGP), retinol, vitamin B12, and folate. Ferritin, folate, and vitamin B12 were measured on the Siemens Immulite 2000 Xpi, while CRP and AGP were measured using the Siemens Dimension Xpand Plus. These biomarkers were analyzed at the Vitamed laboratories in Tashkent. Serum retinol was analyzed using reverse-phase high performance liquid chromatography (LC-20 Prominence with an Auto Sampler, Shimadzu, USA) at ‘MedStandart’ laboratories in Tashkent. Retinol was extracted into acetonitrile, centrifuged, and the supernatant was injected into the system using a Hypersil GOLD aQ Analytical HPLC Column, 3 µm, 4.6 × 100 mm (1202Y28, Thermo Fisher Scientific, Waltham, MA, USA), with a mobile phase of 83% acetonitrile, 0.1% trimethylamine, and 17% water by volume. Retinol was detected at 325 nm using a photo-diode array detector. Retinyl acetate was added as an internal standard before extraction (1716002, United States Pharmacopeia (USP) Reference Standard, Sigma-Aldrich (MERCK), USA). Purified retinol was used to construct the external standard curves (95144, BioXtra, ≥97.5% HPLC, Sigma-Aldrich (MERCK), USA).

Prior to analyzing survey samples, both laboratories underwent a series of external QC rounds conducted by the US Centers for Disease Control and Prevention, followed by capacity building, until laboratory performance was satisfactory. During the analyses of the survey samples, the laboratory conducted rigorous QC that was externally reviewed twice a week. Iron content in flour was assessed in a three-staged approach. All samples underwent qualitative testing using the iron spot test (AACC method 40–40). For samples containing iron, a semi-quantitative test was conducted (INCAP Method IV). Samples pre-determined to have ≥40 ppm iron underwent quantitative analysis using atomic absorption spectrophotometric method as an additional QC measure; there was good agreement between the semi-quantitative and quantitative method (R^2^ = 0.95), although the quantitative results were consistently higher by 1–7 ppm depending on the fortification levels, likely due to the fact that the quantitative analysis also measured intrinsic iron. 

### 2.3. Parameters and Clinical Thresholds

Hemoglobin concentrations were adjusted for smoking status and altitude using World Health Organization (WHO) guidelines [17]. Hemoglobin concentrations <120 g/L were used to classify NPW as anemic [17]. 

Ferritin concentrations were adjusted for inflammation using CRP and AGP values via the method developed by the Biomarkers Reflecting Inflammation and Nutrition Determinants of Anemia (BRINDA) project [18]. Ferritin concentrations <15 μg/L were used as cut-offs for ID in women [19]. For retinol, a cutoff of 0.7 µmol/L was used to define vitamin A deficiency [20]; however, because few women were found to be vitamin A deficient, we instead used vitamin A insufficiency (VAI) defined as serum retinol concentration <1.05 μmol/L [21]. Retinol concentrations were not adjusted for inflammation [22]. FD was defined as folate concentrations <10 nmol/L (<4.4 ng/mL), and vitamin B12 deficiency was identified when levels were <150 pmol/L (<203 ng/L) [23]. Cut-offs for elevated CRP and AGP were >5 mg/ L and >1 g/L, respectively [24]. 

Undernutrition and overnutrition in NPW were assessed using body mass index (BMI; kg/m^2^). Undernutrition was defined as having a BMI less than 18.5, and overweight/obesity was defined as having a BMI of 25.0 or greater [25].

### 2.4. Data Management and Statistical Analysis

Data were collected using Open Data Kit with built in checks and limits to minimize entry errors; additionally, the research team monitored data quality daily from a remote location outside of Uzbekistan. Laboratory data were double-entered using Microsoft Excel 2010. Data analysis was done using SPSS version 22 using the complex survey module. Standardized statistical weights calculated for households and women accounted for the unequal selection probability among the 14 strata. Because all NPW 15–49 years and all pregnant women in selected households were recruited for survey participation, the household sampling weights could be directly applied to each woman included in the survey sample.

Normality of the distribution of continuous data was checked using histograms and calculating skewness and kurtosis. Factors associated with anemia, ID, and FD were identified using bivariate analyses (see Appendix A). All variables associated with a specific outcome in bivariate analyses with *P* < 0.1 were included in a multivariable Poisson regression model after checking for co-linearity. The Poison regression produced adjusted risk ratios which were compared with unadjusted risk ratios calculated using the statistical weights. Variables included in the bivariate analysis were household variables (residence, region, household wealth quintile, household sanitation and access to safe drinking water source); woman’s physiology and nutrition (age, education, cigarette smoking, lactation, underweight, overweight/obesity, iron and folic acid supplement consumption, dietary diversity, consumption of iron- and folic acid-rich foods, additional iron and folic acid intake from wheat flour (as % of Reference Nutrient Intake (RNI)); woman’s micronutrient and inflammatory status (inflammation, iron, folate and vitamin B12 deficiency, vitamin A insufficiency). 

Household socio-economic status was assessed using data on household characteristics and assets. Principal component analysis was used to calculate an index of household wealth, which was subsequently used to classify households into wealth quintiles [26,27]. 

The weekly quantity of flour consumed in each household was calculated from the reported frequency of purchase and quantity usually purchased each time. The number of adult male equivalents (AMEs) in each household was calculated from the household roster information collected during the household interview [28,29]. The AME is the proportion of a young adult male’s energy requirement needed by each age- and sex-specific group. The proportion of household flour consumed by an individual woman was considered equivalent to the proportion of total AME’s in the household represented by that woman. Estimates of daily flour consumption of >500 g/day were excluded from all analyses as physiologically implausible; this corresponded with the 95th centile of wheat flour consumption among Uzbek women in this survey. The calculation of mean flour intake included households reporting not having consumed wheat flour, but they represented only a small proportion and, thus, did not substantially affect the mean.

RNIs for iron and folate in women were obtained from WHO and the Food and Agriculture Organization (FAO) of the United Nations [30]; for iron, an overall bioavailability of 12% was assumed to determine the target RNI. The additional amounts of iron or folic acid coming from household flour were calculated as a fraction of the RNI. Subsequently, %RNI categories were arbitrarily created using thresholds that would result in somewhat similarly sized groups of women with additional intake of iron and folic acid from fortified flour. For iron intake, the following RNI categories were created: 0% RNI (1461 women), 0.1%–39.9% RNI (308 women), and ≥40% RNI (268 women). For folic acid, the RNI categories were: 0% RNI (1093 women), 0.1%–69.9% RNI (265 women), and ≥70% RNI (192 women). Because the folic acid concentration in wheat flour was not directly measured, the folic acid levels were calculated as a proportion of the iron content in the wheat flour (using a 1:33 iron to folic acid ratio as found in the premix used for fortification). 

### 2.5. Ethics and Consent

Ethical approval for the survey was obtained from the Ministry of Health of Republic of Uzbekistan Ethical Committee (Letter 2/12, dated 17 March 2017), as well as from Health Media Lab, an additional external institutional review board appointed by UNICEF (approved on 27 February 2017). Informed verbal consent was sought from the head of the household or, if absent, from the spouse or another adult household member. For individual questionnaires and blood collection, written informed consent was sought from participating women. Confidentiality of information from the respondents was upheld with utmost care throughout data collection, processing, and analysis. Severe cases of anemia were referred to the nearest health facility for follow-up. 

## 3. Results

### 3.1. Household and Demographic Characteristics of Respondents

The household response rate of 94.6% participation in the national survey was very high [12], and of the 2269 enrolled women, more than 97% provided a blood sample. 

Characteristics of households and women are given in Table 1 below. The majority of households resided in rural areas and were male-headed. Access to safe drinking water source and improved sanitation facilities was high in Uzbekistan. About one-third of flour samples were fortified (iron concentration ≥30 ppm). 

The mean age of the participating women was just over 31 years, and the majority had some sort of higher education. Three quarters were married at the time of the survey, one fifth was breastfeeding, and almost half of the women had given birth during the two years prior to the survey. Less than half of women met the criteria for an adequately diverse diet (≥4 food groups consumed in the previous 24 h). Mean wheat flour consumption in women was 210 g/day.

### 3.2. Flour Consumption and Intake of Iron and Folate from Fortified Flour

Wheat flour consumption shows a bimodial-like distribution (Figure 1). About one-third of women consumed no or small amounts of wheat flour, with a drop in women with intermediate consumption (100–200 g/d) followed by a peak in women consuming 200–350 g/d. 

The calculated estimates of additional RNI for iron and folate provided by wheat flour consumption were a combination of the quantities of wheat flour consumed and the concentration of fortificants. These estimates are shown in Figure 2, and show that despite only a relatively small proportion of women not consuming wheat flour (<8%; see Figure 1), 72% of women did not get any additional iron and folic acid from wheat flour. Thus, about two-thirds of surveyed women consumed unfortified wheat flour. Close to a fifth of the women were estimated to consume ≥30% of the RNI for iron coming from wheat flour, while almost a quarter had an additional intake of ≥30% of the RNI for folic acid. Only very few women were estimated to have had an additional iron intake in excess of 100% of the RNI for iron, while this was the case for about 5% of women for folic acid. 

### 3.3. Risk Factors for Anemia, Iron and Folate Deficiencies

Appendix A shows the bivariate analysis of various risk factors and anemia, ID, and FD. Of the factors investigated in the bivariate analysis, the ones associated with anemia, ID, or FD were included in the multivariable Poisson regression model (*P* < 0.1). For anemia: region, wealth, woman’s age, ID and VAI; for ID: region, sanitation, woman’s education, lactation status, underweight, and iron supplement consumption; for FD: residence, region, woman’s age, lactation status, folic acid supplement consumption, iron and folic acid intake from wheat flour (%RNI), and inflammation. 

The resulting multivariable model shows that both ID and VAI were significant risk factors for anemia, while the %RNI of iron additionally consumed was not associated with anemia (Table 2). 

Similarly, the %RNI of iron had little effect on the risk of ID. On the other hand, VAI was a statistically significant risk factor for ID. Women with a higher education were at slightly higher risk for ID. Current breastfeeding was a significant risk factor only in unadjusted bivariate analysis, but no longer in the adjusted model, while being underweight was indicative of a lower risk for ID in the adjusted model but not the bivariate analysis. Of the factors included in the multivariable model, VAI was most strongly related to ID. Although not statistically significant, women with inflammation tended to have a lower prevalence of ID.

In contrast to the models of anemia and ID, women with additional intake of folate from fortified wheat flour had a lower risk of FD than women without additional intake. However, a clear dose-response relationship was absent. Current breastfeeding and inflammation were risk factors for FD (Table 2). 

## 4. Discussion

The findings of this analysis show the potential of a wheat flour fortification program in Uzbekistan to ameliorate folate status among women. Women consuming fortified wheat flour have a lower prevalence of FD. In contrast, these results also show that iron fortification of wheat flour, as practiced prior to this survey, has little potential for decreasing the prevalence of anemia or ID. Women who consume fortified wheat flour did not have a lower prevalence of either anemia or ID. 

In spite of the demonstrated association between consumption of fortified wheat flour and folate status, the effectiveness of the fortification program in ameliorating FD in the population is limited. This is not only reflected in the high prevalence of FD, but also in the large proportion of women (more than 95%) who were found to have folate levels below the cut-off for optimal NTD prevention recommended by the WHO ([33,34]). Poor household coverage with fortified wheat flour, about one-third, limits the population impact of wheat flour fortification. The household coverage of fortified wheat flour must be expanded in order to lower the prevalence of FD among women in Uzbekistan and the risk for NTD in their offspring. 

In contrast, the failure of wheat flour fortification to ameliorate iron status and anemia may be due to factors in excess of poor coverage. Prior to and during the survey, the premix used to fortify wheat flour in Uzbekistan included electrolytic iron, despite the fact that it is well known that electrolytic iron is poorly absorbed. However, current recommendations state that in populations with an average flour consumption of 210 g/d, which was found in Uzbek women, electrolytic iron may be used if more bioavailable forms are not feasible [9]. Nonetheless, Uzbekistan has begun using NaFeEDTA instead of electrolytic iron since this survey. This step, when combined with increasing the coverage of fortified wheat flour, should lead to a decline in the prevalence of ID and, with it, anemia, as ID has been identified as the main anemia risk factor. A recent meta-analysis investigating the effect of large-scale fortification on anemia, iron, and folate deficiency [7], among other outcomes, portrays a relatively consistent pattern of a positive effect of large-scale fortification on anemia, ID, and FD. However, the authors stress that context and implementation factors are important when assessing programmatic sustainability and impact.

Aside from ID, thalassemia, chronic kidney disease, and uterine fibroids have been identified as anemia risk factors among women in Central Asia [35]. It remains to be elucidated to what extent these factors contribute to the overall anemia burden in Uzbekistan. 

The findings presented here are subject to certain limitations. Because the data analyzed were collected in a cross-sectional survey, we cannot make strong conclusions regarding causality. However, the slight suggestion of a dose-response relationship, and the relative strength of the association between fortified wheat flour consumption and folate status may indicate potential causality. 

To estimate iron and folic acid intake of Uzbek women, we used the AME approach, which has been described by FAO and others [29] and has been extensively used in the recent past to estimate macronutrient intake. On the other hand, it has been used less frequently to estimate micronutrient intake. In this study, we used this approach to estimate the quantity of wheat flour consumed by participating women. To estimate micronutrient intake, we used the measured iron concentrations of household wheat flour samples. However, there are inherent limitations to the AME approach. It cannot include foods consumed outside the household, nor does it account for within-household food-sharing inequalities. Women may not routinely consume a proportion of a fortified food staple equivalent to their contribution to total household AMEs due to the cultural or societal preference given to certain family members. These limitations may potentially lead to an underestimation of nutrient intake from wheat flour, which may mask a potential association between iron intake and ID. However, despite such a potential underestimation, the effect of increased folic acid intake on FD was easily detectable in this study. 

We used RNI rather than the estimated average requirement (EAR), because no EARs have been provided by the WHO for iron in women 15–49 years [23]. Although EARs exist for folic acid, we chose to work with RNI for both micronutrients, to be consistent. Also, we used the same thresholds for lactating women and non-lactating women to determine iron and folate requirements. The median breastfeeding duration in our data was 21 months, and only about one-third of children from lactating women were less than 6 months old. Hence the term lactating defines a very heterogeneous group, ranging from a high additional nutrient requirement for intensely breastfeeding women to much lower additional needs for mothers with only occasional breastfeeding. Thus, using different RNI cut-offs might have led to an over-or underestimation of the RNI depending on the breastfeeding intensity. 

Further, we used serum folate to assess folate status and not red blood cell folate, which better reflects folate status as it is less prone to day to day intake variations. Notwithstanding, the WHO considers both red blood cell and serum folate as suitable biomarkers to evaluate the impact of public health interventions with folic acid [34].

## 5. Conclusions

While the wheat flour fortification program did not provide large amounts of iron and folic acid to the majority of women of reproductive age in Uzbekistan and did not reduce the risk of anemia and ID, there is a possible dose-response relationship for FD and folic acid intake. These results indicate that the fortification program has the potential for an impact, but requires higher coverage of adequately fortified wheat flour and a more bioavailable iron fortificant. The latter point has recently been addressed by the government by switching from electrolytic iron to NaFeEDTA.

## Figures and Tables

**Figure 1 nutrients-12-00714-f001:**
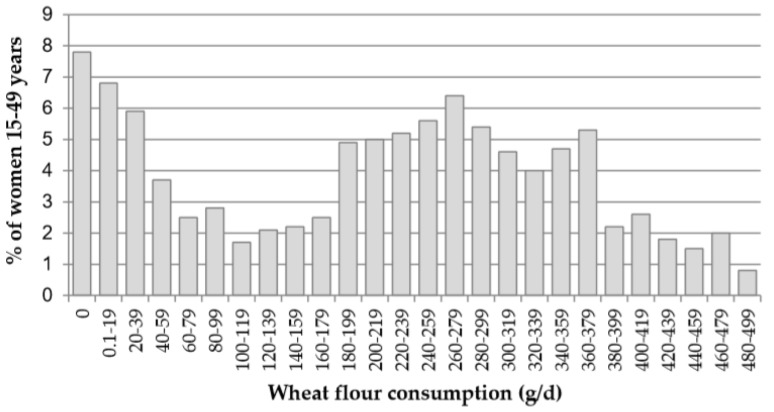
Distribution of wheat flour consumption among non-pregnant women (NPW) in g/d, Uzbekistan 2017.

**Figure 2 nutrients-12-00714-f002:**
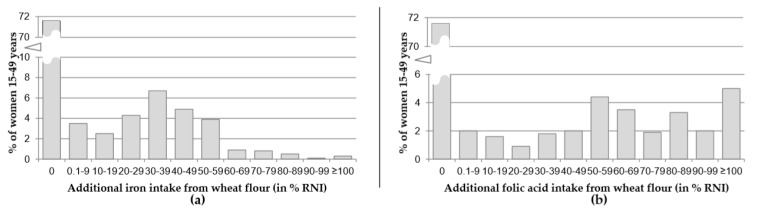
Additional iron (**a**) and folic acid (**b**) intake from fortified wheat flour for NPW, expressed as % of the reference nutrient intake (RNI), Uzbekistan 2017.

**Table 1 nutrients-12-00714-t001:** Household and demographic characteristics of respondents, Uzbekistan 2017.

Variable	Number with Characteristic (N)	% or Mean ^a^	(95% CI) ^b^
**Household characteristics**			
Household size, mean	1806	6.0	(5.8, 6.2)
Urban residence, %	505	28.2%	(23.9, 33.0)
Male headed household, %	1452	79.3%	(76.4, 81.9)
Safe drinking water ^c^, %	1802	99.9%	(98.9, 100.0)
Improved sanitary sanitation ^c^, %	1600	89.2%	(86.9, 91.0)
Fortified wheat flour ^d^, %	517	29.2%	(26.1, 32.5)
**Non-pregnant women (NPW) characteristics**			
Age (years), mean	1806	31.2	(30.8, 31.6)
Higher education ^e^, %	1041	58.4%	(55.5, 61.2)
Currently married, %	1329	73.1%	(70.8, 75.2)
Currently breastfeeding, %	341	19.1%	(17.1, 21.3)
Given birth in past 2 years, %	872	49.5%	(47.0, 52.1)
Number of prior pregnancies, mean	1806	2.3	(2.2, 2.4)
Meeting minimum dietary diversity ^f^, %	716	40.6%	(37.7, 43.5)
Consumption of predominant type of flour (g/d), mean	1806	210	(199, 221)

Note: The N’s are un-weighted numbers for each subgroup. ^a^ Percentages or means are weighted for unequal probability of selection, except distribution of regions. ^b^ CI = confidence interval, calculated taking into account the complex sampling design. ^c^ Definitions as per [31]. ^d^ Fortified wheat flour was defined as flour with an iron content of ≥15 ppm. ^e^ Higher education defined as having attended more than 11 years of school (special secondary or higher levels). ^f^ As recommended by Food and Agriculture Organization of the United Nations (FAO) and and FHI 360 [32].

**Table 2 nutrients-12-00714-t002:** Unadjusted and adjusted relative risks (RRs) for anemia and iron and folate deficiency (FD) in non-pregnant women 15–49 years of age, Uzbekistan 2017.

Characteristic	Category	Unadjusted (Bivariate Analysis)		Adjusted (Poisson Regression)	
		Relative Risk	95% CI	Relative Risk	95% CI
**Anemia ^a^ (*N* = 1473)**					
Iron intake from fortified flour (as %RNI)	≥40%	1.30	(0.96, 1.70)	1.20	(0.91, 1.50)
	0.1–39.9%	1.10	(0.87, 1.43)	1.20	(0.93, 1.40)
	0%	referent	-	referent	-
Iron deficiency ^b^ (ID)	Yes	4.24	(3.15, 5.70)	4.70	(3.50, 6.50)
	No	referent	-	referent	-
Vitamin A insufficiency ^c^ (VAI)	Yes	1.89	(1.52, 2.34)	1.50	(1.30, 1.90)
	No	referent	-	referent	-
**ID (*N* = 1473)**					
Iron intake from fortified flour (as %RNI)	≥40%	0.97	(0.81, 1.20)	1.00	(0.86, 1.20)
	0.1–39.9%	0.90	(0.76, 1.06)	0.91	(0.78, 1.05)
	0%	referent	-	referent	-
Educational level ^d^	Secondary or less	0.86	(0.77, 0.95)	0.85	(0.77, 0.94)
	Special secondary or more	referent	-	referent	-
Currently breastfeeding	Yes	1.15	(1.02, 1.30)	1.10	(0.99, 1.20)
	No	referent	-	referent	-
Underweight (BMI < 18.5)	Yes	0.81	(0.62, 1.05)	0.74	(0.58, 0.96)
	No	referent	-	referent	-
VAI	Yes	1.45	(1.30, 1.61)	1.40	(1.30, 1.60)
	No	referent	-	referent	-
Any inflammation ^e^	Yes	0.88	(0.75, 1.03)	0.86	(0.74, 1.01)
	No	referent	-	referent	-
**FD ^f^ (*N* = 1550)**					
Folic acid from fortified flour (as %RNI)	≥70%	0.62	(0.49, 0.79)	0.67	(0.53, 0.85)
	0.1–69.9%	0.70	(0.58, 0.86)	0.74	(0.62, 0.88)
	0%	referent	-	referent	-
Currently breastfeeding	Yes	1.27	(1.13, 1.43)	1.20	(1.10, 1.40)
	No	referent	-	referent	-
Any inflammation	Yes	1.23	(1.07, 1.42)	1.20	(1.03, 1.30)
	No	referent	-	referent	-

^a^ Anemia defined as hemoglobin < 120g/L; ^b^ ID defined as serum ferritin < 15µg/L; ^c^ VAI defined as retinol < 1.05 µmol/L; ^d^ Higher education defined as having attended more than 11 years of school (special secondary or higher levels); ^e^ Any inflammation defined as elevated C-reactive protein (CRP) (>5mg/L) and or elevated alphal-acid-glycoprotein (AGP) (>1g/L). ^f^ folate deficency (FD) defined as serum folate <10 nmol/L.

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
