# Peer review of "Risk Factors for Anemia and Micronutrient Deficiencies among Women of Reproductive Age—The Impact of the Wheat Flour Fortification Program in Uzbekistan"

_nutrients, 2020, doi:10.3390/nu12030714_

Round 1

Reviewer 1 Report

The article is well written, and the discussion is well supported by the methods and results, giving due consideration to aspects related to actual reach of fortified flour to the intended population, problems related to bioavailability of micronutrients, and the potential role of confounders such as inflammation and breastfeeding.

However, I would like to bring up to the authors’ attention two important caveats, with the hope they may be addressed to make a more solid contribution to the literature.

The first one relates to the method to evaluate folate status. As the authors correctly identify (Rogers et al, 2018), nationally representative prevalence estimates for folate deficiency are scarce, particularly in low income countries. Even more scarce are data based on the determination of folate in red blood cells using a harmonized microbiological assay, as recommended by the World Health Organization (World Health Organization. 2015. Serum and red blood cell folate concentrations for assessing folate status in populations. Vitamin and Mineral Nutrition Information Systems. http://www.who.int/vmins/en/). Folate in red blood cells is the best indicator to assess long-term folate status, while serum folate concentration is more a reflection of recent dietary intake. The use of different laboratory methods makes it difficult to compare results amongst countries, again as has been highlighted by Rogers et al (2018).

The second caveat refers to the use the cutoff recommended by the World Health Organization to identify the level of folate insufficiency above which the maximum protection against the risk of a neural tube defect (NTD) pregnancy is found. This cutoff identifies a red blood cell folate concentration of 906 nmol/L at the population level as the optimum concentration to protect women of reproductive age against a folate-sensitive NTD. Concentrations below this cutoff identify folate insufficiency, related to higher risk of NTDs; this cutoff is much higher than the one used to identify folate deficiency, set at 305 nmol/L, which is used to identify a high risk of megaloblastic anemia. A recent paper has attempted to identify plasma folate concentrations associated with the red blood cell folate concentration for optimal NTD prevention (Meng-Yu et al. Am J Clin Nutr 2019; 109:1452-1461), using data collected over 10 years ago in the context of an RCT in China. The correlation between these two measures is not free from its own caveats (e.g., responses are different according to BMI, genotype, and low plasma vitamin B12 status), but it may provide some guidance on the plasma threshold to use. These authors found that, for nonanemic, non-vitamin B-12 deficient women, a cutoff of 25.5 nmol/L was associated with the optimal NTD prevention, setting this at 8 NTD per 10,000 births, similar to the protective effect conferred by a red-blood cell folate concentration of 906 mol/L.

In summary, if the authors find a way to address these concerns, they will be in a better position to contribute with their research to the important body of knowledge generated to support large-scale food fortification programs in the world.

Author Response

We thank reviewer 1 very much for the very helpful comments.

Regarding comment 1 we have added a sentence to the limitation section of the discussion (lines 361-364) pointing out that RBC folate is the more suitable biomarker, but that according to WHO RBC as well as serum folate are suitable to assess the performance of public health interventions.

Regarding comment 2: We have re-run the analyses using the cut-off for folate insufficiency as recommended by Meng. Using this cut-off, more than 95% of women were classified as insufficient (see Table below).

wFolDef Deficient is <25.5 nmol/L (11.30 ng/ml)

Estimate

95% Confidence Interval

Unweighted Count

Lower

Upper

% of Total

1 Deficient

95.2%

93.9%

96.2%

1988

2 Nl

4.8%

3.8%

6.1%

103

Total

100.0%

100.0%

100.0%

2091

We then tried to repeat the Poisson regression model with the new cut-off as the outcome variable. We included those variables for which the p value in the new bivariate table was less than or equal to 0.10. This model included minimum dietary diversity, consumed folate rich foods in the past day, percent RNI iron intake from fortified flour, and percent RNI folate intake from fortified flour. However, the model would not run, since folate insufficiency is so common that the number of women without folate insufficiency is insufficient to handle multiple-variable modelling. Given the very high prevalence of insufficiency and the recommendation by Meng that the cut-off of 25.5nmol/l should not be used in population with high prevalence of B12 insufficiency (about 34% in Uzbekistan) we did not present results using that cut-off in the results section, but added a sentence to the discussion section (lines 308-9).

Reviewer 2 Report

     General comments:

  • Overall, the manuscript is very important with considerable implications for fortification policy and programming in Uzbekistan.
  • There are many grammatical edits that need to be made throughout the manuscript. Please update accordingly. There are also some acronyms that are not introduced in the main paper which need to be defined at first use (e.g. ID and UNS).

  1. Introduction:
  • The introduction focuses largely on folate deficiency, yet the focus on the article goes beyond this micronutrient deficiency. Please add more relevant information about anemia, iron deficiency, and other micronutrient deficiencies that are the focus of these analyses and which are important for the country.
  • Lines 52-54: There is reference to a substantial decrease in coverage of fortified flour from 2008 to 2017. Please explain this decrease more in terms of whether this was related to policy, implementation of a fortification program, whether this varied geographically in the country, etc.
  • Lines 56-57: Please include the amounts of folic acid, zinc, etc. that are used for fortification similar to how the amount of iron is indicated.
  • Lines 62-63: Please specify which micronutrient deficiencies this analysis will look at.
  • Lines 62-64: The abstract mentions vitamin A insufficiency, yet vitamin A is not mentioned in the objectives. Please add vitamin A insufficiency or clarify why this is not listed. Vitamin B 12 was also measured so consider also adding this micronutrient here as well. It could be useful to make it clear in this section and in the methods which variables are outcomes compared with which micronutrients are used as risk factor variables in analyses (e.g. vitamin A).
  • Line 64: This sentence states “non-pregnant 15-49 years old.” Please add “women.” Moreover, in the methods there is discussion of the study including all women and does not list pregnant women as part of any exclusion criteria. Please clarify in this objectives statement whether pregnant women were also included. In the abstract, there is also reference to breastfeeding. In the objectives, please also clarify whether breastfeeding women were included in analyses as they have different nutritional requirements.
  • More information about sources of the micronutrients of interest (iron and folic acid) aside from fortified flour will be useful to include here (e.g. other foods commonly consumed, their consumption rates if known, supplementation programs/supplementation practices, etc.).

  1. Materials and Methods:
  • Lines 71-72: This sentence states that non-pregnant women, pregnant women, and children were recruited in the survey. Please clarify who is included in the analyses in this manuscript and what the exclusion criteria were. It is unclear whether pregnant women were excluded, whether children are included, and whether breastfeeding women were included.
  • There is no final sample size included in the methods. Please add the final sample size of women that were included in analyses and clarify whether all were non-pregnant and age 15-49 years. Please also specify if these women were a subset of the larger survey population and the response rate. This is important because often when venous blood samples are taken, response rates can be variable which can impact the representativeness of the data. If women recruited for biological sample collection (i.e. venous blood) represent a subset of the larger survey population, please specify selection criteria for this subsample and provide justification of its representativeness.
  • Lines 94-95: There is mention of dietary habits collected. Please specify what dietary habits were collected and what “habits” means. Were quantitative dietary intake recalls undertaken, food frequency questionnaires, etc.? Were food group data collected for all foods that the women consumed? Were data only collected on the consumption of fortified and fortifiable foods and not consumption of other foods? Specific information on what dietary intake data were collected needs to be added.
  • Were any questions asked about supplementation or is supplementation common in Uzbekistan? I understand the introduction states that folic acid supplementation during pregnancy starts late and is limited. Were questions asked about iron or IFA supplementation? Was supplementation considered in analyses and why was it not controlled for?
  • Lines 96-97: Please specify whether anthropometric measurements were taken for all women in addition to children/who anthropometric measurements were taken on. The reference for standard anthropometry methods is for children, yet the target population for analyses is children.
  • Lines 98-100: Please specify whether blood was taken on all participating women, including pregnant women. Please also include the sample size of participants who participated in blood collection.
  • Lines 147-149: “Chronic energy deficiency” is outdated terminology. Moreover, low BMI (less than 18.5) can be due to more than just energy deficiency. Reference 25 is outdated. The current terminology for a BMI of less than 18.5 among adults is “underweight” according to WHO. Please update the terminology and reference.
  • Line 156: Here “eligible women” are referenced, yet the eligibility criteria are not mentioned. Please specify the eligibility criteria.
  • Line 160: Please list the factors associated with anemia, iron deficiency, and folate deficiency that were identified.
  • Lines 168-169: Please justify why the frequency and quantity of flour purchase was used as a proxy for consumption at the household and individual level with individual estimates calculated using the AME. Is there evidence showing that household purchasing is associated with women’s flour intake? If so, please include this. While there is information about this in the discussion, it will be important to include some of these details here as well.
  • Lines 168-178: The calculation of the proportion of household flour consumed by an individual woman assumes that distribution of flour/flour-based foods is equivalent and based on sex/age within the household. Please justify or explain this assumption in this study’s context. These calculations also assume that the women meet their energy requirement daily. Is there evidence to support this? Do some women exceed or consume less than their energy requirement? If so, this is a limitation of the calculation that should be discussed.
  • Lines 177-178: Please quantify the proportion of households that did not consume wheat flour. While it is listed as small here, Figure 1 shows that the percent of women consuming no wheat flour is not insubstantial.
  • Lines 182-189: Please add more information about the calculation and justification of the RNI categories and thresholds for iron and folic acid. Moreover, line 183 states that % RNI categories were created using thresholds that would result in similarly sized groups, but the groups listed are of substantially different sizes (i.e. groups range from 192 to 1461 women).
  • Please explain why folic acid concentrations in flour were not measured since that was a primary outcome of interest and listed as a micronutrient in fortified flour in the country.

  1. Results:
  • Figure 2: It looks like the women who did not consume wheat flour were included in Figure 2. It looks like the percent of women who did not consume wheat flour is about 8%. Were there any differences in the Figure 2 estimates without women who did not consume wheat flour? Why were women who did not consume wheat flour included in this analysis/figure?
  • Were RNIs adjusted for non-pregnant women who were breastfeeding? Nutrient intake requirements and recommendations are different for lactating women. Since 19% of women in this study were breastfeeding, it will be important to indicate how these women were treated in analyses and what, if any, adjustments were made. If adjustments were not made, please explain why.
  • Figure 2: Please clarify how the additional folic acid intake from flour was quantified if folic acid concentrations were not measured in the flour samples.
  • Table 2: Please be consistent with the use of significant digits.
  • Lines 252-255, Table 2: Table 2 shows that currently breastfeeding and any inflammation variables were significant in the folate deficiency models. This is not explained in the text. Please add explanations about these associations in the text.

  1. Discussion:
  • It could be useful to include information about the assumed concentrations of folic acid in the fortified flour since it was not measured.
  • It could be useful to provide information about other sources of iron and folic acid in Uzbekistan to prevent deficiency in addition to fortification for a comprehensive picture (e.g. other foods, supplements).
  • It could be interesting to explore or hypothesize about other causes of anemia in Uzbekistan based on evidence in this section as there are causes aside from micronutrient deficiencies that may be interesting to share more information about since there could be implications and expectations for what fortified flour can do for anemia.
  • There is no reference to the breastfeeding women in the study population in the discussion. Since these women represented 19% of the study population, it is important to explain why including them in the analyses with non-breastfeeding women was appropriate or to cite this as a limitation when interpreting findings. The argument could also be made that analyses included breastfeeding women to be conservative since their needs exceed the non-pregnant women normal population, but this should be made clear.
  • The discussion and conclusion could reference the success of other countries that have iron and folic acid flour fortification programs as examples of the effectiveness of this intervention on micronutrient deficiencies when coverage is high.

Author Response

We thank the reviewer for the very helpful comments!

There are many grammatical edits that need to be made throughout the manuscript. Please update accordingly. There are also some acronyms that are not introduced in the main paper which need to be defined at first use (e.g. ID and UNS).

We have thoroughly reviewed the manuscript again for grammar and style, as well as have we made sure to use already introduced acronyms more consistently.

  1. Introduction:

The introduction focuses largely on folate deficiency, yet the focus on the article goes beyond this micronutrient deficiency. Please add more relevant information about anemia, iron deficiency, and other micronutrient deficiencies that are the focus of these analyses and which are important for the country.

Information on the prevalence of anemia and ID has been added to the introduction.

Lines 52-54: There is reference to a substantial decrease in coverage of fortified flour from 2008 to 2017. Please explain this decrease more in terms of whether this was related to policy, implementation of a fortification program, whether this varied geographically in the country, etc.

We do not have sufficient data to explore and substantiate potential explanation for the decrease. Thus, such explanation would be speculative. 

Lines 56-57: Please include the amounts of folic acid, zinc, etc. that are used for fortification similar to how the amount of iron is indicated.

Added.

Lines 62-63: Please specify which micronutrient deficiencies this analysis will look at.

The study investigates anemia, ID and FD and their underlying risk factors. This is now clarified in lines 62-64.

Lines 62-64: The abstract mentions vitamin A insufficiency, yet vitamin A is not mentioned in the objectives. Please add vitamin A insufficiency or clarify why this is not listed. Vitamin B 12 was also measured so consider also adding this micronutrient here as well. It could be useful to make it clear in this section and in the methods which variables are outcomes compared with which micronutrients are used as risk factor variables in analyses (e.g. vitamin A).

The objective of the study was to assess risk factors for anemia, ID and FD, but not vitamin A insufficiency or vitamin B12 deficiency, since these two micronutrients were not in the fortification premix at the time of the survey. However, Vitamin A insufficiency has been identified as a risk factor and is therefore mentioned in the abstract. Lines 62-64 have been reworded accordingly. All variables included in the model are presented in supplementary table 1.

Line 64: This sentence states “non-pregnant 15-49 years old.” Please add “women.” Moreover, in the methods there is discussion of the study including all women and does not list pregnant women as part of any exclusion criteria. Please clarify in this objectives statement whether pregnant women were also included. In the abstract, there is also reference to breastfeeding. In the objectives, please also clarify whether breastfeeding women were included in analyses as they have different nutritional requirements.

We have modified the manuscript to clarify this point. With regard to breastfeeding women, we have clarified this in the method section (lines 73-4). As to the inclusion of lactation status in the analysis, please refer to a response to a comment further down.

More information about sources of the micronutrients of interest (iron and folic acid) aside from fortified flour will be useful to include here (e.g. other foods commonly consumed, their consumption rates if known, supplementation programs/supplementation practices, etc.).

In order to keep the introduction concise and due to the lack of appropriate or up-to-date data prior this survey, we decided not to present data on other sources of micronutrients of interest in the introduction. You will see though that in the revised method section and from supplementary table 1 that we indeed considered nutritional supplements and consumption of iron- and folic acid-rich foods in our analysis.

  1. Materials and Methods:

Lines 71-72: This sentence states that non-pregnant women, pregnant women, and children were recruited in the survey. Please clarify who is included in the analyses in this manuscript and what the exclusion criteria were. It is unclear whether pregnant women were excluded, whether children are included, and whether breastfeeding women were included.

This has been clarified.

There is no final sample size included in the methods. Please add the final sample size of women that were included in analyses and clarify whether all were non-pregnant and age 15-49 years. Please also specify if these women were a subset of the larger survey population and the response rate. This is important because often when venous blood samples are taken, response rates can be variable which can impact the representativeness of the data. If women recruited for biological sample collection (i.e. venous blood) represent a subset of the larger survey population, please specify selection criteria for this subsample and provide justification of its representativeness.

All non-pregnant women who were included in the survey and provided a blood sample were included in the analysis. Response rate was > 90%. This has been clarified at the beginning of the results section. Given the high response rate, we don’t think further explanations are required to ascertain the reader of a minimal survey population bias.

Lines 94-95: There is mention of dietary habits collected. Please specify what dietary habits were collected and what “habits” means. Were quantitative dietary intake recalls undertaken, food frequency questionnaires, etc.? Were food group data collected for all foods that the women consumed? Were data only collected on the consumption of fortified and fortifiable foods and not consumption of other foods? Specific information on what dietary intake data were collected needs to be added.

Minimum dietary diversity (24h recall) and the consumption of and knowledge about fortified and fortifiable foods and consumption of vitamin and mineral supplements has been assessed. This is now clarified in lines 98-100.

Were any questions asked about supplementation or is supplementation common in Uzbekistan? I understand the introduction states that folic acid supplementation during pregnancy starts late and is limited. Were questions asked about iron or IFA supplementation? Was supplementation considered in analyses and why was it not controlled for?

Consumption of supplements was assessed and included in the model (see supplementary table 1). It was not included in the multivariate model since there was no significant association with folate deficiency. We have clarified this in lines 175-81 of the methods section as well.

Lines 96-97: Please specify whether anthropometric measurements were taken for all women in addition to children/who anthropometric measurements were taken on. The reference for standard anthropometry methods is for children, yet the target population for analyses is women.

Anthropometric measurements were taken for all participants. This is now clarified and simplified: although the survey included children and pregnant women, the analysis presented in this manuscript only focuses on non-pregnant women. As such, we have removed mentioning of other population groups.

Lines 98-100: Please specify whether blood was taken on all participating women, including pregnant women. Please also include the sample size of participants who participated in blood collection.

A blood sample was collected from all non-pregnant women and children 6-59months of age. For pregnant women only on-site hemoglobin analyses was conducted and no blood samples was collected for later analyses. This is now clarified in the methods section. As mentioned in the previous comment, we have removed mentioning of population groups other than the non-pregnant women from this manuscript.

Lines 147-149: “Chronic energy deficiency” is outdated terminology. Moreover, low BMI (less than 18.5) can be due to more than just energy deficiency. Reference 25 is outdated. The current terminology for a BMI of less than 18.5 among adults is “underweight” according to WHO. Please update the terminology and reference.

We updated the terminology and reference to the following: WHO STEPS Surveillance Manual: The WHO STEPwise approach to non-communicable disease risk factor surveillance, pages 3-5-8 and 3-5-9. World Health Organization 2017. Geneva, Switzerland. (found at: https://www.who.int/ncds/surveillance/steps/STEPS_Manual.pdf)

Line 156: Here “eligible women” are referenced, yet the eligibility criteria are not mentioned. Please specify the eligibility criteria.

Done in the methods section, we think this is clear now.

Line 160: Please list the factors associated with anemia, iron deficiency, and folate deficiency that were identified.

All variables included in the bivariate analyses are presented in supplementary table 1 including p-values. Those variables associated with a specific outcome in bivariate analyses with P < 0.1 were included in a multi-variable Poisson regression model after checking for co-linearity. We agree with the reviewer that this information should also be presented in the main body of the manuscript and modified the manuscript accordingly (175-81 and lines 269-274).

Lines 168-169: Please justify why the frequency and quantity of flour purchase was used as a proxy for consumption at the household and individual level with individual estimates calculated using the AME. Is there evidence showing that household purchasing is associated with women’s flour intake? If so, please include this. While there is information about this in the discussion, it will be important to include some of these details here as well.

While literature on the validation of the exact procedure used here is scarce, there is emerging literature confirming the usefulness of the AME approach, when carefully calculated and provided that household composition data is available and in-house consumption is the predominant form of consumption (Bermudez 2012, Adams 2019). Our data fulfill these criteria for women of reproductive age and due to the availability of the exact household composition, our analysis is more accurate than could be obtained from an analysis of a household-income and expenditure survey (HIES). Thus, while we agree that indirect measures of intake tend to be less accurate compared to the direct measures, direct methods to assess dietary intake are often too costly and complex to be rolled out on a national level and have so far rarely been used for national food and nutrition policy making. The main limitation of the approach we used is that it does not capture food eaten outside the house. This is outlined in the discussion. 

Lines 168-178: The calculation of the proportion of household flour consumed by an individual woman assumes that distribution of flour/flour-based foods is equivalent and based on sex/age within the household. Please justify or explain this assumption in this study’s context. These calculations also assume that the women meet their energy requirement daily. Is there evidence to support this? Do some women exceed or consume less than their energy requirement? If so, this is a limitation of the calculation that should be discussed.

As already mentioned above, the approach we use might not yield a 100% accurate estimation of the individual intake, but the approach has been previously used in other settings and despite some limitations, bears one advantage over single-pass 24 h recall: it is less prone to day-day variation. Repeated 24 h recall assessments or weighed food records would be more accurate, but not feasible at large scale. 

Lines 177-178: Please quantify the proportion of households that did not consume wheat flour. While it is listed as small here, Figure 1 shows that the percent of women consuming no wheat flour is not insubstantial.

Because we used HH approach calculating the AME, the proportion of HH not consuming is similar to the proportion of women.  We agree that 8% not consuming wheat flour is not insubstantial and changed the text from “very small” to “relatively small”. As a matter of fact, the proportion of women from households without flour consumption compared to the proportion of women consuming unfortified flour is about 1/8 which we think can considered be ‘small’.

Lines 182-189: Please add more information about the calculation and justification of the RNI categories and thresholds for iron and folic acid. Moreover, line 183 states that % RNI categories were created using thresholds that would result in similarly sized groups, but the groups listed are of substantially different sizes (i.e. groups range from 192 to 1461 women).

We agree with the reviewer that the groups have different sizes, but all those women who consumed unfortified wheat flour and thus had an additional RNI intake of 0% from wheat flour had to be grouped together regardless of the grouping criterion. RNI categories were chosen in a way that groups of women with additional intake of folic acid and iron had similar sizes. We have modified the text accordingly.  

Please explain why folic acid concentrations in flour were not measured since that was a primary outcome of interest and listed as a micronutrient in fortified flour in the country.

Folic acid is added to wheat flour as part of a premix containing several micronutrients. It is a valid procedure to measure only one of the added stable micronutrients (if added as a premix, which is the case in Uzbekistan), to draw conclusions on the concentration of the other micronutrients in the premix. Further, measuring folic acid in flour is an expensive and complicated analysis for which no laboratory with a quality control track-record could be identified in Uzbekistan (and exportation was not an option). We considered it safer to use the ratio than using folic acid results that cannot be trusted.

Results:

Figure 2: It looks like the women who did not consume wheat flour were included in Figure 2. It looks like the percent of women who did not consume wheat flour is about 8%. Were there any differences in the Figure 2 estimates without women who did not consume wheat flour? Why were women who did not consume wheat flour included in this analysis/figure?

We included all women with serum folate results in the analysis. This is why we did not exclude them from figures 1 and 2. Excluding them would not change the overall picture and we think that from the public health relevance point of few all women included in the survey should be considered. We changed the wording in the results section, so that it is clearer to the reader that all women have been included in the analyses.   

Were RNIs adjusted for non-pregnant women who were breastfeeding? Nutrient intake requirements and recommendations are different for lactating women. Since 19% of women in this study were breastfeeding, it will be important to indicate how these women were treated in analyses and what, if any, adjustments were made. If adjustments were not made, please explain why.

We thank the reviewer for this thoughtful comment. Indeed, while we did include lactating women in the analysis, we did not use a different RNI cutoff for this group. Instead, lactation status was included in the model as a covariate. Based on this reviewer’s comments, we looked at the specific data in more detail and although we did run the model using two different cutoffs, we are not certain that accounting for lactation over and above including it in the multivariate model is warranted:

The metabolic demand of breastfeeding declines with the age of the infant because complementary feeding contributes an increasing proportion of the child’s nutrient requirements as the child ages. Thus, a woman exclusively breastfeeding a 6-month old infant has a very different nutrient drain than a woman providing occasional comfort breastfeeding for a 20-month old toddler.  Our definition of currently lactating is based on the question “Are you currently breastfeeding?” without additional specification about the frequency/quantity. This includes any breastfeeding in the past 24 hours (similar to the definition of breastfeeding for children) and thus, there will be many mothers who are providing minimal nutrition to their breastfeeding child. The median duration of breastfeeding in Uzbekistan was 21 months, so many children 6 months of age and older were reported as breastfeeding. The table below is for children, not women, but it demonstrates that less than one-third of breastfeeding children were less than 6 months of age, the age range which might be expected to provide the greatest nutrient drain on the mother. So, if we assign the lactating RNI to all mothers who report lactating, we will be substantially underestimating the RNI for iron while overestimating the RNI for folate.  

AgeGrp6Mos 6-month age groups * BFCurrent Combination of CBF3 (still BF) and CBF4 (BF 24 hrs) Crosstabulation

BFCurrent Combination of CBF3 (still BF) and CBF4 (BF 24 hrs)

Total

1 Yes

2 No

AgeGrp6Mos 6-month age groups

1 <6

Count

240

5

245

% within BFCurrent Combination of CBF3 (still BF) and CBF4 (BF 24 hrs)

30.3%

0.4%

11.4%

2 6-11

Count

199

12

211

% within BFCurrent Combination of CBF3 (still BF) and CBF4 (BF 24 hrs)

25.2%

0.9%

9.8%

3 12-17

Count

193

43

236

% within BFCurrent Combination of CBF3 (still BF) and CBF4 (BF 24 hrs)

24.4%

3.2%

11.0%

4 18-23

Count

108

129

237

% within BFCurrent Combination of CBF3 (still BF) and CBF4 (BF 24 hrs)

13.7%

9.5%

11.0%

5 24-29

Count

29

206

235

% within BFCurrent Combination of CBF3 (still BF) and CBF4 (BF 24 hrs)

3.7%

15.2%

11.0%

6 30-35

Count

6

179

185

% within BFCurrent Combination of CBF3 (still BF) and CBF4 (BF 24 hrs)

0.8%

13.2%

8.6%

7 36-41

Count

8

232

240

% within BFCurrent Combination of CBF3 (still BF) and CBF4 (BF 24 hrs)

1.0%

17.1%

11.2%

8 42-47

Count

4

183

187

% within BFCurrent Combination of CBF3 (still BF) and CBF4 (BF 24 hrs)

0.5%

13.5%

8.7%

9 48-53

Count

2

213

215

% within BFCurrent Combination of CBF3 (still BF) and CBF4 (BF 24 hrs)

0.3%

15.7%

10.0%

10 54-59

Count

2

152

154

% within BFCurrent Combination of CBF3 (still BF) and CBF4 (BF 24 hrs)

0.3%

11.2%

7.2%

Total

Count

791

1354

2145

% within BFCurrent Combination of CBF3 (still BF) and CBF4 (BF 24 hrs)

100.0%

100.0%

100.0%

Thus, based on the response above, we have not modified the tables and results, but have added a short section in the discussion section to address this point; see lines 366 ff.

Figure 2: Please clarify how the additional folic acid intake from flour was quantified if folic acid concentrations were not measured in the flour samples.

It is mentioned in the method section that “Because the folic acid concentration in wheat flour was not directly measured, the folic acid levels were calculated as a proportion of the iron content in the wheat flour (using a 1:33 iron to folic acid ratio as found in the premix used for fortification).”

Table 2: Please be consistent with the use of significant digits.

This has been corrected

Lines 252-255, Table 2: Table 2 shows that currently breastfeeding and any inflammation variables were significant in the folate deficiency models. This is not explained in the text. Please add explanations about these associations in the text.

The results are presented in the text, see lines 288-89.

Discussion:

It could be useful to include information about the assumed concentrations of folic acid in the fortified flour since it was not measured.

The assumed concentration is already reported in the manuscript. Folic acid is relatively stable to heat and humidity; thus, premixes, baked products, and cereal flours, retain almost 100 percent of the added folic acid after six months of storage (https://www.dsm.com/content/dam/dsm/nip/en_US/documents/stability.pdf.)

It could be useful to provide information about other sources of iron and folic acid in Uzbekistan to prevent deficiency in addition to fortification for a comprehensive picture (e.g. other foods, supplements).

As previously mentioned in a comment on the introductory section, for the sake of conciseness and due to the lack of appropriate or up-to-date data prior this survey, no discussion on other data sources is provided. However, because we included a series of variables in the survey, such as consumption of nutritional supplements and consumption of iron- and folic acid-rich foods, these variables were included in the analysis. As such, this information is included in supplementary table 1. However, because none of these variables were statistically significantly associated with any of the outcome variables, they were not included in the multivariable model. And as such, no further discussion about their potential effect is warranted for this manuscript.

It could be interesting to explore or hypothesize about other causes of anemia in Uzbekistan based on evidence in this section as there are causes aside from micronutrient deficiencies that may be interesting to share more information about since there could be implications and expectations for what fortified flour can do for anemia.

A paragraph has been added to the discussion (lines 327-9).

There is no reference to the breastfeeding women in the study population in the discussion. Since these women represented 19% of the study population, it is important to explain why including them in the analyses with non-breastfeeding women was appropriate or to cite this as a limitation when interpreting findings. The argument could also be made that analyses included breastfeeding women to be conservative since their needs exceed the non-pregnant women normal population, but this should be made clear.

Please refer to above response, we have inserted a short paragraph in the discussion section. Over and above what was previously stated, please keep in mind that micronutrient requirements of lactating women do not universally exceed those of non-lactating women. For example, the RNI for iron in a population whose diet provides 12% bioavailability is about 25 mg per day for non-lactating women and only 12 mg per day for lactating women. Although this is only one micronutrient, this represents one of two micronutrients we analyze in detail.

Moreover, from a programmatic point, when targeting interventions to non-pregnant women, nutrition program managers generally do not distinguish between lactating and non-lactating women. And just to highlight the aforementioned, “lactating women” is not a homogeneous group; women exclusively breast-feeding a five-month-old infant have a much greater nutrient burden than women providing comfort breast-feeding for a toddler. We were unable find in any set of recommendations or biomedical literature a definition distinguishing lactating women with a high burden from lactating women with a low burden. Therefore, we question the usefulness to distinguish “lactating” women from non-lactating women because really, lactating women with older children should not be treated as different from non-lactating women. To justify this choice, we added a short paragraph in the discussion section, see lines 370-7.

The discussion and conclusion could reference the success of other countries that have iron and folic acid flour fortification programs as examples of the effectiveness of this intervention on micronutrient deficiencies when coverage is high.

Although we have already provided a short update on the current status of the literature on this topic (see lines 46-50), we have added the following statement to re-iterate the promise of large-scale food fortification programs (lines 323-26):

A recent meta-analysis investigating the effect of large-scale fortification on – among other outcomes – anemia, iron and folate deficiency (Keats 2019) portrays a relative consistent pattern of a positive effect of large-scale fortification on all three outcomes. However, authors stress that ‘context and implementation factors are important when assessing programmatic sustainability and impact .’

Reviewer 3 Report

This is an informative and unbiased assessment of fortification. The data quality appears to be truly meaningful and will truly contribute to the knowledge pool in the relevant food fortification applications.

Authors recommends relatively more iron fortification. However, the critical safety concerns of iron have not been adequately discussed. Therefore, authors should clarify the iron deficiency and supplementation difficulty concerns as well as the critical safety issues of excess iron intake including accidental ingestion.

Author Response

We do thank reviewer 3 for the positive overall feedback on our manuscript. With regard to the comments about safety of iron interventions, while this is a concern indeed, in particular in areas of the world with high inflammation-pressure, e.g. malaria-endemic countries, or in countries with a high prevalence of hemoglobinopathies that may lead to hematochromatosis. None of these conditions are highly prevalent in Uzbekistan. Our data suggest that while overall, a third of women had some sort of inflammation, only about 3% are at the onset of the inflammatory cascade (CRP only elevated) and about 10% showed signs of an acute inflammatory response (both CRP and AGP elevated). As such, we think it is safe to assume that the inflammation pressure in Uzbekistan is rather low. Therefore, we don’t think that in this context, the addition of a paragraph will provide clarity to the manuscript.